# Visible-Light Active Photocatalytic Dual Layer Hollow Fiber (DLHF) Membrane and Its Potential in Mitigating the Detrimental Effects of Bisphenol A in Water

**DOI:** 10.3390/membranes10020032

**Published:** 2020-02-21

**Authors:** Roziana Kamaludin, Zatilfarihiah Rasdi, Mohd Hafiz Dzarfan Othman, Siti Hamimah Sheikh Abdul Kadir, Noor Shafina Mohd Nor, Jesmine Khan, Wan Nor I’zzah Wan Mohamad Zain, Ahmad Fauzi Ismail, Mukhlis A Rahman, Juhana Jaafar

**Affiliations:** 1Advanced Membrane Technology Research Centre (AMTEC), Universiti Teknologi Malaysia, Skudai 81310, Malaysia; roziana.kamaludin7@gmail.com (R.K.); afauzi@utm.my (A.F.I.); mukhlis@petroleum.utm.my (M.A.R.); juhana@petroleum.utm.my (J.J.); 2Institute of Medical Molecular Biotechnology, Faculty of Medicine, Sungai Buloh Campus, Universiti Teknologi MARA (UiTM), Jalan Hospital, Sungai Buloh 47000, Malaysia; zatilfarihiah27@gmail.com; 3Biochemistry and Molecular Medicine Department, Faculty of Medicine, Sungai Buloh Campus, Universiti Teknologi MARA (UiTM), Jalan Hospital, Sungai Buloh 47000, Malaysia; jesminek@salam.uitm.edu.my (J.K.); wnizzah@salam.uitm.edu.my (W.N.I.W.M.Z.); 4Institute for Pathology, Laboratory and Forensic Medicine (I-PPerForM), Universiti Teknologi MARA (UiTM), Sungai Buloh 47000, Malaysia; drshafina@salam.uitm.edu.my; 5Pediatric Department, Faculty of Medicine, Sungai Buloh Campus, Universiti Teknologi MARA (UiTM), Jalan Hospital, Sungai Buloh 47000, Malaysia

**Keywords:** bisphenol A, photocatalytic, dual layer hollow fiber, visible-light driven, in-vivo

## Abstract

The presence of bisphenol A (BPA) in various water sources has potentially led to numerous adverse effects in human such as increased in blood pressure and derangement in liver function. Thus, a reliable treatment for the removing BPA is highly required. This present work aimed to study the efficiency of visible light driven photocatalytic dual-layer hollow fiber (DLHF) membrane for the removal of BPA from water and further investigated its detrimental effects by using an in-vivo model. The prepared membranes were characterized for their morphology, particles distribution, surface roughness, crystallinity and light absorption spectra. The removal of 81.6% and 86.7% in BPA concentration was achieved for N-doped TiO_2_ DLHF after 360 min of visible and UV light irradiation, respectively. No significant changes for all three groups were observed in liver function test meanwhile the rats-exposed to untreated BPA water shows significance blood pressure increment contrary to rats-exposed to treated BPA water. Similarly, the normal morphology in both jejunum and ileum were altered in rats-exposed to untreated BPA water group. Altogether, the presence of N-doped TiO_2_ in DLHF are shown to significantly enhance the photocatalytic degradation activity under visible irradiation, which effectively mitigates the effect of BPA in an in-vivo model.

## 1. Introduction

Bisphenol A (2,2-bis(4-hydroxyphenyl)propane) is a ubiquitous endocrine disrupting compound used in the manufacturing of epoxy resins and polycarbonate plastics articles [1]. BPA is commonly used as liners in food and drinks packaging [2], various plastic materials and toys, as well as appearing in numerous types of electronic equipment [3]. Exposure to BPA has been associated with many adverse health effects such as decrease in sperm production [4], preimplantation of zygotes for embryo development [5], thyroid function disruption [6], obesity [7], diabetes [8] and many more. Continuous exposure to BPA should be a greater concern to pregnant women and their fetuses as numerous reported BPA effects are more detrimental in them [9]. BPA threats to human health have led to the rising interest in developing a strategies to reduce the exposure to BPA and removing the BPA harmful effects as well before it could be consumed.

Recently, the development of heterogeneous visible light active (VLA) photocatalysts has captured much attention recently due to their nature of easy recycling and simple synthesis method. Interestingly, it exhibited high reactivity under visible light irradiation, and hence had greater benefit for photocatalysis application. In recent years, nitrogen-doping TiO_2_ (N-doped TiO_2_) has gained intense interest due the broad absorption in the visible region the increased light absorption, hence efficient photoactivation by solar light. Previously, N-TiO_2_ from calcined products of TiCl_4_ in NH_4_OH was reported with photocatalytic high visible light activity which was ascribed to the interstitial nitrogen atoms within TiO_2_ lattice units [10]. In another research, visible light active N-doped TiO_2_ was prepared by mixing urea and TiO_2_ powder under microwave irradiation. The presence of N-doped makes N-doped TiO_2_ active under visible light irradiation with reduced band gap to 2.9 eV. It was reported that the prepared N-doped TiO_2_ had excellent photocatalytic activity with complete the degradation of RR4 after 60 min [11]. In addition, conformal thin films of nitrogen doped TiO_2_ on membranes have been fabricated of via atomic layer deposition (ALD) resulting in a lowered band gap with enhanced permeance and photocatalytic activity for degrading organic pollutants under solar irradiation [12]. Recently, an N-doped TiO_2_ photocatayst was successfully synthesized through acid modified sol-gel method [13]. The performance evaluation proved that the resultant N-doped TiO_2_ exhibited good optical properties with excellent degradation of BPA and RB5 dyes up to 90% under visible-light irradiation.

In the meantime, the immobilization of photocatalyst into membrane matrix not only resolved the separation problems faced by photocatalytic slurry system type but simplifies the overall photocatalytic process. In this regard, the membrane has the simultaneous task of supporting the immobilized photocatalyst as well as acting as a molecular separation barrier for treatment process [14] while the presence of a photocatalyst enhance the rate of the chemical reaction under light irradiation [15]. Therefore, degradation reaction and separation processes were performed in a single unit. Interestingly, the incorporation of high loading of photocatalyst on the outer membrane layer surface will ensure the fast degradation of contaminant. Similarly, Ooi [16] mentioned this configuration offers various advantages for the chemical synthesis includes simple preparation technique due to inexpensive catalyst molecules, fewer toxicity issues and cost saving. Most importantly, the greatest benefit is the ease of removal of the catalyst from the reaction waste in addition to no risk of contamination in the final products. Interestingly, the study on the use of small organic molecules as catalysts covers various modes of catalysis, including common Brønsted and Lewis acid−base catalysis, nucleophilic catalysis, redox catalysis, organometallic catalysis, enzymatic catalysis, and photochemical catalysis. Few new studies have worked on the immobilization of catalyst into membrane matrix. Cao et al., [17] reported on yeast-immobilized catalytically active membrane using immersion phase inversion technique through coating a porous yeast/PES layer on a PDMS pervaporation membrane. Interestingly, the catalytically active membrane exhibited comparable fermentation activity as free yeasts and maintained similar pervaporation performance in comparison with traditional PDMS membrane in addition to a robust durability of repeated fermentation-pervaporation coupling experiments. In the meantime, Zhao, Zhang and Wang, [18] have demonstrated the AuNP-immobilized BCP membranes with excellent catalytic performance as high as 100% in the continuous flow catalytic reduction. To date, no studies have been reported on the degradation of BPA from contaminated water by visible light-driven photocatalytic DLHF membrane. To the very best of our knowledge, most of the previous works focused on the degradation of BPA using photocatalytic degradation under UV light exposure. It is well known that a structure of BPA is made up of strong bonds hence, very difficult to degrade by conventional treatment methods. Hence, the present study is conducted to investigate the efficiency of DLHF membrane for photocatalytic degradation of BPA in contaminated water under visible light irradiation. In this present work, visible light active nitrogen-doping (TiO_2_) was constructed via simple sol-gel method followed by the incorporation of the photocatalyst into DLHF membrane by using co-extrusion phase inversion method. The presence of VLA N-doped TiO_2_ on the outer surface of DLHF membrane would offer great benefit for the utilization of visible light irradiation. The membranes morphology was characterized by scanning electron microscopy (SEM), energy dispersion X-ray analysis (EDX) as well as atomic force microscopy (AFM). The photocatalytic activity of the developed visible light-driven DLHF membrane was assessed via submerged membrane photo reactor with comparison of commercial TiO_2_ P25 DLHF membrane. The toxicity effect of BPA was measured quantitatively by liver function test (LFT) including total bilirubin, albumin, total protein and alkaline phosphatase level. The blood pressure (BP) was recorded using volume pressure recording (VPR) sensor technology during early and end of the treatment. The effect of BPA on the major organ of BPA exposure, SI was observed qualitatively by haematoxylin and eosin (H& E) staining technique. The outcome of this study will outline the potential of visible light-driven photocatalytic DLHF membranes as a promising technology for treating BPA in contaminated water and reduce its side effects for better environment and better healthcare.

## 2. Methodology

### 2.1. Materials

#### 2.1.1. Photocatalytic Dual Layer Hollow Fiber (DLHF) Membrane

The nitrogen-doped (N-doped) TiO_2_ nanoparticles prepared previously [13] and commercial Degussa TiO_2_ P25 purchased from Sigma Aldrich were used as photocatalysts. Polyvinylidene fluoride (PVDF, Kynar 760, Colombes, France), Polyethelene glycol (PEG) 6000, and dimethylacetamide (DMAC, QReC) were of analytical reagent grade and were used as received without further dilution.

#### 2.1.2. In-Vivo Models

All chemical for in-vivo work were purchased from Sigma Aldrich, except as otherwise stated. Tween-80 was used as vehicle control of treatment (with 0.4% of Tween-80 in total solution). For Haematoxylin and Eosin (H&E) standard protocols, formalin (Labchem), paraffin, absolute ethanol (Merck, Darmstadt, Germany.), xylene and Haematoxylin and Eosin dyes (Fisher Scientific, Waltham, MA, USA) were purchased and used as received.

### 2.2. Fabrication of Photocatalytic Dual Layer Hollow Fiber Membrane

Photocatalytic DLHF membranes were fabricated using dry/wet co-spinning technique as reported elsewhere [19,20,21]. The dope solution was prepared accordingly as reported previously [21]. Dope composition and spinning conditions was summarized in the Table 1.

Subsequently, the fabricated membranes were soaked into tap water at room temperature (RT) for 24 h to remove the residual solvent followed by treatment in ethanol: water, 50:50 for 1 h and 100% of ethanol for another 1 h. For each treatment, 50 pieces of fiber segments of 30 cm length was put together to form a bundle. Four to six bundles were treated at a time in 3L of water or solvent, respectively. Finally, the fibers membranes were dried at RT for 72 h. The fiber bundles were stored in plastic cover until further used.

### 2.3. Membrane Properties Analysis

The morphology of photocatalytic DLHF membranes were examined by scanning electron microscopy (SEM; Model: TM 3000, Hitachi, Tokyo, Japan) as well as energy dispersion Xray analysis (EDX; X-MaxN 51-XMX1011, Oxford Instrument, Abinton, UK) to investigated the presence of N-doped TiO_2_ and TiO_2_-P25 on membrane surface. The measurement of crystallinity and phase formation of the sample was carried out by X-ray diffractometer (XRD, Model: D5000, SIEMENS, Munich, Germany) at 40 kV and 40 mA, which employed a CuKα radiation at a wavelength of 0.15418 nm at an angular incidence of 2θ = 20–80°. FTIR spectra for DLHF sample was performed with Perkin Elmer FT-IR attenuated total reflection (ATR) spectrophotometer and diamond ATR sampling accessory. The spectrum of the sample was scanned with the wave number ranging from 650 to 4000 cm^−1^. Contact angle of the fiber samples were measured using the contact angle goniometer (Model: OCA 15EC, Dataphysic, Filderstadt, Germany) which equipped with image processing software to evaluate the degree of hydrophilicity. Tensile strength of the hollow fibers was established by LRX, LLYOD test machine. At least six fibres with 50 mm length were tested with a load cell of 2.5 kN, at a constant elongation velocity of 10 mm/m at room temperature. Different fiber of the same batch was used for the measurement and the average data ± SD were presented. The optical absorption of the photocatalytic DLHF sample was measured with UV-Vis-NIR spectrophotometer (UV-3101PC Shidmadzu, Kyoto, Japan).

### 2.4. Photocatalytic Activity Evaluation

The degradation efficiency of the photocatalytic N-doped TiO_2_ DLHF in comparison with the commercial TiO_2_-P25 immobilized into DLHF membrane was evaluate by using submerged membrane photoreactor [21]. BPA with initial concentration of 5 ppm (BPA > 99%; Sigma-Aldrich, St. Louis, MO, USA) was used as contaminant model. Prior to light irradiation, the membrane modules were immersed into the beaker and BPA solution was oxygenated without light irradiation for 120 min to achieve adsorption/desorption equilibrium. Then 10 mL of aliquot was collected and treated as the initial concentration (C_o_) of BPA. Subsequently, the solution was irradiated under visible/UV light. 10mL of treated aliquots was collected at 30 min interval (C_t_) within 360 min of experiment period. The concentration different of BPA in each samples were measured by High Performance Liquid Chromatography (HPLC) analysis (Agilent Technologies 1260 Affinity, Santa Clara, CA, USA) coupled with UV detector at 280 nm. The percentage of removal efficiency is calculated by using the simple equation below:$$Degradation\;of\;BPA=\frac{Co-Ct}{Ct}x\;100$$

The untreated BPA water (feed) and treated BPA water (permeate) were stored in a BPA-free container for further used on in-vivo models.

### 2.5. Animal Care and BPA Exposure

The in-vivo work was carried out using female Sprague-Dawley strain of rats with the approval of Animal Research and Ethics (UiTM CARE) Committee for animal experiments (UiTM CARE: 254/2018(3/8/2018)).The committee confirming that all experiments were performed in accordance with relevant guidelines and regulations. Female Sprague–Dawley rats were purchased, aged between 150–180 days and randomly divided into 3 groups: Group 1 (Vehicle Control Group (VHC); Tween 80 control group), Group 2 (Untreated (UT) BPA water from Section 2.5), and Group 3 (Treated (T) BPA water from Section 2.5) (*n* = 4–6). Rats were housed individually with free access food and water under standard conditions (22 °C, 12 h light-dark cycle), and were administered with BPA water for 21 days. Control rats were given water containing 1% Tween 80, the concentration was used as a vehicle for BPA solution. Water bottles and caged made of BPA-free components were used for all of these studies to avoid potential contamination of BPA from sources.

#### 2.5.1. Dissection

At day 21, dissection was performed with anesthesia by intraperitoneal injection of pentobarbital at 50 mg/kg body weight/dose. Jejunum and ileum of rats from each group were collected to evaluate the toxicity of VHC and BPA-exposed rats qualitatively by Haematoxylin and Eosin (H&E) staining technique. The difference in weight and water (treatment) intake of rats from each group were also recorded.

#### 2.5.2. Liver Function Test

Blood of the rats were collected and stored in ethylenediaminetetra-acetic acid (EDTA) vacuum blood collection tube. The tube was centrifuged at 3500× *g* for 15 min to separate serum and total blood. Prior to liver function test (LFT), all serum were stored at −20 °C to avoid any degradation on the blood samples. The LFT included total bilirubin, albumin, total protein and alkaline phosphatase level.

#### 2.5.3. Blood Pressure

Volume pressure recording (VPR) sensor technology was used to conduct the non-invasive blood pressure (BP) measurement of all groups of rats. Each rat was placed in a strainer with a proper size to allow comfortable breathing and avoid stress. VPR system were measured the blood flow and blood volume in the tail thus generated BP readings simultaneously. Systolic BP (SBP) is a top number that refers to amount of pressure in arteries during contractions of heart muscles while diastolic BP (DBP) is a bottom number refers to the beating of heart muscles.

#### 2.5.4. Heamatoxylin and Eosin (H&E) Staining

Jejunum and ileum were collected, weighed and fixed in 10% neutral buffered formalin, processed in the tissue processing machine for 24 h and subsequently embedded with paraffin wax. Wax blocks were cooled and sectioned at a thickness of 4 µm followed by incubation at 37 °C overnight for 24 h for Heamatoxylin and Eosin (H&E) staining. The staining was conducted following the standard protocol of H&E. The paraffin sections were, drying on a hot plate, and then immersing into xylene followed by dehydration by absolute ethanol, graded series of ethanol and rinsed with running tap water. Slides were stained with Haematoxylin dye (Thermo Fisher Scientific, Waltham, MA, USA) followed by counterstain in Eosin (Thermo Fisher Scientific). The sections were then dehydrated in an absolute alcohol and xylene followed by mounted in DPX mountant (Sigma Aldrich) and covered with glass cover slips. Slides were examined using a light microscope (Olympus BX51, Melville, NY, USA) microscope prior to digital analysis using Zeiss LSM510META laser-scanning confocal microscope.

## 3. Results and Discussion

### 3.1. Physical Properties of DLHF Membranes

Previously, Kamaludin et al., [21] has successfully developed a dual layer hollow fibre (DLHF) membrane immobilized with visible light active N-doped TiO_2_ for photocatalytic activity. The spun visible-light driven N-doped TiO_2_ DLHF was best fabricated at ratio N-doped TiO_2_/PVDF was 0.5. To date, there is no commercial N-doped TiO_2_ available and to the very best of our knowledge, there is no study on N-doped TiO_2_ DLHF. Therefore the comparison study between N-doped TiO_2_ with commercial TiO_2_-P25 would be a great help to assess the capability and the benefit of the newly fabricated membrane. For comparison, internationally commercial TiO_2_-P25 was successfully incorporated into dual layer hollow fiber membrane at the same ratio of TiO_2_-P25/PVDF. The morphological nature of both DLHF membranes was shown in Figure 1. Both membranes possessed sandwich-like structure without any distinctive difference being observed apart from their photocatalyst particles. There are no delamination occurs and both layers are compatible with each other due to the mutual diffusion of both inner and outer layers [21].

Figure 2 represent the image from EDX mapping. The outer layer thickness of approximately 8.12 and 8.37 μm (Figure 2b1,b2, respectively) was produced for N-doped TiO_2_ and TiO_2_-P25 DLHF respectively when flowrate of 2 mL/min was used in this work. It is also interesting to note that the outer layer thickness might also depending on the photocatalyst loadings as it will influence the dope solution viscosity. Our previous study [13] have reported that N-doped TiO_2_ had slightly crystallite size as compared to the commercial TiO_2_-P25, hence at the same dope composition, the N-doped TiO_2_ DLHF dope solution have less particles loadings than TiO_2_-P25 DLHF doped solution. Less viscose dope solutions has higher tendency to flow quickly, resulting in the formation of thinner outer layer. Similar results were obtained in previous work [22] which reported the outer layer thickness of 8.74 μm of the DLHF membranes was obtained with flowrate of 2 mL/min was used during the fabrication process.

Tensile strength and elongation at break, contact angle, water flux, porosity and band gap obtained for both membranes were summarized in Table 2. The tensile strength and elongation at break of TiO_2_-P25/PVDF DLHF are slightly higher than N-doped TiO_2_ DLHF. This might due to its thicker outer layer, hence stronger structure as compared to single layer hollow fiber membrane (SLHF) reported in previous finding [23]. Interestingly, the presence of N-doped TiO_2_ and TiO_2_-P25 particle at the outer membrane layer has improved the membrane hydrophilicity properties. In comparison, the fabrication of nonwoven membrane supports from bamboo fiber reinforced poly(lactic acid) composites resulted in increased mechanical stability from 32.7 to 73.3 MPa while the elongation at break remained virtually the same [24]. In the meantime, TRIP-TR-460-30 PBO membrane recently fabricated displayed good mechanical properties with tensile strength, and elongation at break of 58 MPa and 4.3%, respectively [25]. Similarly, three PI-TBs membranes fabricated for gas separation application had tensile strengths in the range of 52.9–114.2 MPa, and elongation at break of 8.9%–10.8% [26]. Even though the tensile strength and elongation at break of the fabricated DLHF in this current study was low however, as a fair comparison, the membrane composition, polymer material and application needed to be considered. Besides that, the dispersion of N-doped TiO_2_ and TiO_2_-P25 is one of the most important factors in the fabrication of the DLHF membranes as it impacts directly on membrane adsorption capacity and photocatalytic activity. Both membranes are composed of a microporous structure with relatively large pores as determined SEM images (Figure 1a3,b3) and supported by porosimeter analysis (Table 2). The porous inner layer structure of both DLHF was achieved with the addition of the pore former, PEG 6000 into the dope solution. The porosity results of both DLHF membranes are in good agreement with pure water flux. However, the pure water flux result might be lower as compared to the single layer hollow fiber (made of PVDF and TiO_2_) reported in previous work, which around 67.9 L/m^2^.h. This might due to the slower diffusion of water through the hydrophobic inner layer (PVDF) of the DLHF membranes [22]. In the meantime, the obtained band gap energy for N-doped TiO_2_/PVDF and TiO_2_-P25/PVDF DLHF membrane are found to be 2.64 and 2.9 eV, respectively. The incorporation of N-doped TiO_2_ nanoparticles onto outer membrane surface significantly improved the absorbance capability. The absorption edge of the optical response does shift to the red region of the spectrum upon N-doping which subsequently narrowing the band-gap hence improved the absorbance capability under visible irradiation [27]. This narrow band gap of N-doped TiO_2_ DLHF strongly enhanced their photocatalytic properties under visible light region [21].

In the meantime, the XRD patterns of the N-doped TiO_2_ and TiO_2_-P25/PVDF DLHF membranes were shown in Figure 3. The characteristic peaks of TiO_2_ P25 and N-doped TiO_2_ crystalline at 2θ = 25.4° and 2θ = 27.5° (Figure 3a) were predominant in N-doped TiO_2_ and TiO_2_-P25/PVDF DLHF membranes (Figure 3b). It could be seen that the XRD pattern of N-doped TiO_2_ DLHF membranes had three crystalline characteristic peaks at 2θ = 25.4° and 2θ = 27.5° and 38.7° that was analogous with the predominant characteristic peaks of N-doped TiO_2_/TiO_2_ P25 and PVDF membrane, respectively. Similarly, TiO_2_-P25/PVDF DLHF shows the same crystalline characteristic peaks, respectively. These characteristic peaks indicated that there are interactions between polymer and N-doped TiO_2_/TiO_2_ P25 which influenced the PVDF crystal structure (transition of α to β phase) in the DLHF membranes. The direct evidence of polymorph change of the PVDF membrane can also be seen where the neat PVDF powder exhibited peaks at 2 of 18.69° and 20.11° (Figure 3b); characteristic of α-polymorph, which was close to the previous reported study [28].

FTIR spectra of fabricated N-doped TiO_2_/PVDF DLHF and N-doped TiO_2_ powder were shown in Figure 4. These spectra present the relationship between the fabricated membranes and the N-doped TiO_2_ nanoparticles. The FTIR spectra were recorded in the range 400–4000 cm^−1^. The band between 400 and 800 cm^−1^ were assigned to the strong vibration of Ti–O and Ti–O–Ti species. The peaks at 1346–1417 and 1050–1095 cm^−1^ could be ascribed to the nitrogen atoms embedded in the TiO_2_ network [29,30]. These results clearly demonstrated that the nitrogen doped TiO_2_ has been incorporated into the DLHF structure. The presence of the visible light active photocatalyst in the membrane configurations will enhance the photo absorption capacity under the visible light region [11]. The intense band at 1180 cm^−1^ was assigned to CF_2_ bond meanwhile, the vibration observed at 2900 cm^−1^ belonged to the OH species. The peaks of 765, 796, and 976 could be assigned to the typical peaks of α-phase PVDF crystals as reported by previous study [31].

### 3.2. Photocatalytic Degradation Evaluations

The degradation of BPA in water by N-doped TiO_2_ DLHF in comparison with TiO_2_-P25 DLHF was evaluated under visible and UV light irradiation. Figure 5 demonstrate the photocatalytic degradation curves of BPA under visible and UV light irradiation in the presence of N-doped TiO_2_ and TiO_2_-P25 DLHF membrane. As can be seen in the both figures, the photolysis study proved that the BPA is stable under both visible light and UV irradiation with significantly low degradation percentage of less than 1% which indicated that the degradation process can only occur in the present of photocatalytic membrane under the light irradiation. Figure 5a showed that the photocatalytic N-doped TiO_2_ DLHF exhibited excellent photocatalytic behavior to degrade the BPA with degradation activity of 81.6% after 360 min of visible irradiation. Low photocatalytic activity of TiO_2_-P25 DLHF with degradation of 21.7% was recorded after 360 min of reaction. In the meantime, as can be seen from Figure 5b, under UV irradiation, the N-doped TiO_2_ DLHF had photocatalytic activity with degradation percent of 86.7% after 360 min of irradiation. Meanwhile 90.9% of BPA degradation was observed after 360 min of experiment in the presence of TiO_2_-P25 DLHF. Similar trend in degradation of phenol was reported previously [32]. Those findings also clearly indicated that the photocatalytic N-doped TiO_2_ DLHF was active under UV and visible light irradiation, while TiO_2_-P25 DLHF only active under UV light irradiation due to its large band gap. The presence of visible light-active N-doped TiO_2_ in N-doped TiO_2_ DLHF significantly enhanced the absorption capability hence; promote excellent photocatalytic degradation under visible irradiation.

### 3.3. BPA-Treated Water Ameliorated Its Detrimental Effects in Comparison to BPA-Untreated Water

Rats were randomly divided into 3 groups: Group 1 (Vehicle Control Group (VHC); Tween 80 control group), Group 2 (Untreated (UT) BPA water), and Group 3 (Treated (T) BPA water) (*n* = 4–6). Rats were administered with BPA untreated and treated water by N-doped TiO_2_ DLHF membrane (from Section 2.5) for 21 days. The initial concentration of untreated BPA water was 5ppm [33,34,35]. The effects of BPA on general health status of treated rats were shown in Figure 6. All groups of rats showed no significant weight gained and drinking pattern from treatment day 2 (TD2) to treatment day 21 (TD21), (Figure 6a,b). These results demonstrated that BPA has no effects on the general health of rats even after chronic exposure. Similarly, Desai et al. [36] also reported that the average water consumption was similar in BPA and control groups (BPA = 47.4 ± 3.0 mL/day; Control = 46.4 ± 3.7 mL/day). This finding is in an agreement with previously reported study where they reported no clinical symptoms of intoxication after BPA administration. Behavior, appetite, and increments in body weight of animals treated with BPA were similar to those in the control animals [37]. In contrast, BPA-exposed increase defensive aggression in male Sprague–Dawley rat offspring which prenatally exposed to BPA at 40 μg/kg/dose/day. A similar finding was reported in male CD-1 mouse offspring exposed to BPA at 2 μg/kg/dose/day perinatally [38].

Liver is the primary organ responsible for BPA metabolism, thus knowledge on the toxic effect of BPA on the liver is important. Low dose of BPA could be vulnerable as free BPA circulates throughout the body [39]. Figure 7 shows the LFT results for total bilirubin, albumin, total protein and alkaline phosphatase level of the treated rats. Upon observation, there were no significant differences between control and BPA-treated groups. The LFT results further suggested that the level of BPA in this study was not hepatotoxic. Nevertheless, there was a study demonstrated the associations of urinary BPA with abnormalities of liver function [40].

### 3.4. Blood Pressure (BP) Readings

BP readings of the treated rats showed significant increased values from the first to third week of treatment (Figure 8). The difference in systolic blood pressure (SBP) (Figure 8a) was found to be significance between groups (*p* = 0.013). The results indicated that SBP significantly increased during treatment in untreated group compared to the control group (*p* < 0.05). Interestingly, in the treated BPA group, the SBP readings was decreased from first to third trimester, thus suggested that low level of BPA may not trigger hypertension. When the BPA-exposed groups were compared with the control group, the SBP (in the third trimester) was raised up from 129.25 mmHg in control rats to 146.50 mmHg and 144.67 mmHg in filtered and unfiltered group of rats, respectively (*p* = 0.029). However, for diastolic blood pressure (DBP) (Figure 8b), the difference was not significant (*p* > 0.05), although the readings did increase steadily from first to third week. During the last week of respective treatment, the DBP readings of filtered and unfiltered groups were increased as compared to the control group from 99 to 110.50 mmHg and 115.00 mmHg, respectively (*p* = 0.292). There were a few studies which reported the association of BPA exposure with the development of hypertension [40,41,42]. Hypertension or known as high blood pressure occurs when there are greater forces of blood flows in blood vessels. Those findings suggested that BPA exposure may increases the risk of cardiovascular disease (CVD) that is mediated by raised of blood pressure readings. Based on the results observed in this study, it is in line with few other studies that reported on the prevalence of hypertension in BPA-exposed subjects. Bae et al. [41] found an increment of hypertension prevalence among Korean adults whose were exposed to BPA. This was in agreement with another study conducted in 2015, where they detected a small increase in SBP of female-exposed to BPA at dosage of 5.0 µg/kg/day [43]. Furthermore, a randomized clinical trial conducted previously [36] demonstrated an increment in SBP by 4.5 mmHg after being exposed to BPA. Surprisingly, we also found significant increment in SBP by 15.42. mmHg of untreated BPA group (5 ppm BPA) compared to the control group, indicated that exposure of BPA influenced the BP changes. Although DBP readings were not significant, however the value was higher than normal diastolic BP of SD rat which is 91 mmHg, showed that there is a possibility of developing hypertension [44].

### 3.5. Changes in Morphology of Jejunum and Ileum

In here, H&E staining showed that BPA alters the normal morphology of jejunum and ileum compared to the control group (VHC) which had normal finger-like projections of the villi (Figure 9). Normal projections of villi were disrupted and lost their shape in untreated BPA-exposed group. Interestingly, our treated BPA-exposed group showed similar shape of villi projection as observed in the VHC group. Similarly, previous study has shown that several pathological changes like necrosis, degenerative changes in villus in small intestine tissues of BPA-treated animals [45]. BPA was given at 1/25 of oral LD_50_ and none of the rats died during the experimental period. The microscopic examinations of the rat small intestine showed that BPA induced necrosis and edema which might be due to oxidative damages of BPA on cells [46]. Another study [37] was done to determine the influence of low and high dosage BPA (LD; 0.05mg/kgbodyweight/day, HD; 0.5mg/kgbodyweight/day) on the porcine ileum. Histopathological examination showed that the intestinal and villi structure was preserved. However, in the lumen of intestinal crypts of HD group, the number of eosinophils was higher with visible inflammatory cells than the control and LD groups. Although there is a lot of study about general toxic effects of BPA, there is limited study about the toxicity of BPA on the small intestine.

## 4. Conclusions

The photocatalytic DLHF membranes were successfully fabricated via co-spinning phase inversion method. It has been proved that there was good compatibility between both outer and inner layer due to a strong interfacial interaction between layers. The resultant N-doped TiO_2_ exhibited good optical properties with obtained band gap of 2.64 eV with excellent photocatalytic activity in the presence of ultraviolet and visible irradiation. In addition, the evaluation of BPA removal shows that both DLHF membranes had excellent photocatalytic activity up to 90% of removal under UV light irradiation meanwhile N-doped TiO_2_ DLHF exhibited excellent performance with 81.6% of removal under visible light irradiation. Interestingly, exposure to BPA was shown to increase the incidence of hypertension which may then raise the possibility of other disease development. Furthermore, BPA exposure leads to altered morphology of finger-like projections of villi in the jejunum and ileum. Most importantly, our treated BPA-exposed group showed similar results as observed in the VHC group which indicated that BPA degraded by photocatalytic (DLHF) membrane system effectively mitigates the effect on BPA on the jejunum and ileum.

## Figures and Tables

**Figure 1 membranes-10-00032-f001:**
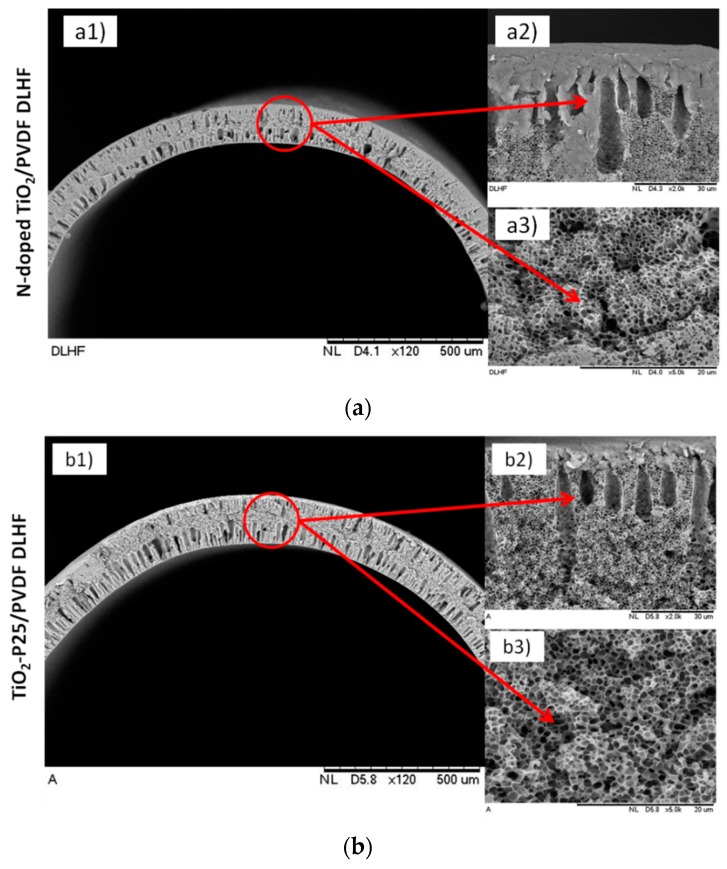
SEM cross-section morphological analysis of DLHF membranes; (**a**) N-doped TiO_2_ DLHF and (**b**) TiO_2_-P25 DLHF (a1 and b1 represent the cross-sectional morphology, a2 and b2 represent the finger like morphology and a3 and b3 represent the sponge-like structure).

**Figure 2 membranes-10-00032-f002:**
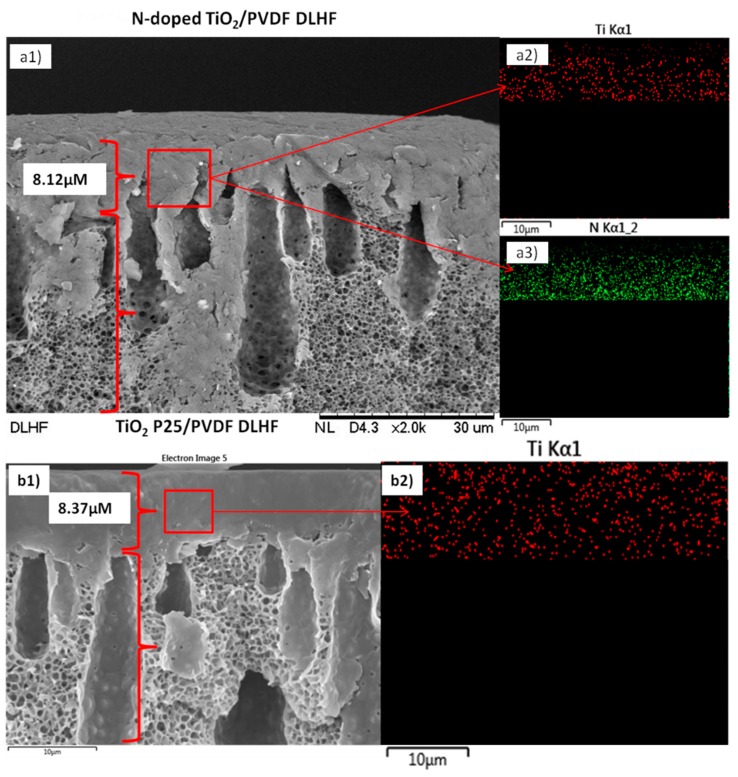
SEM images and energy dispersion X-ray analysis (EDX) mapping of (**a1**,**a2**,**a3**) N-doped TiO_2_ and (**b1**,**b2**) TiO_2_-P25 DLHF membranes ((**a2**,**b2**) represent the mapping image of Ti element, and a3 represent the mapping image of N element).

**Figure 3 membranes-10-00032-f003:**
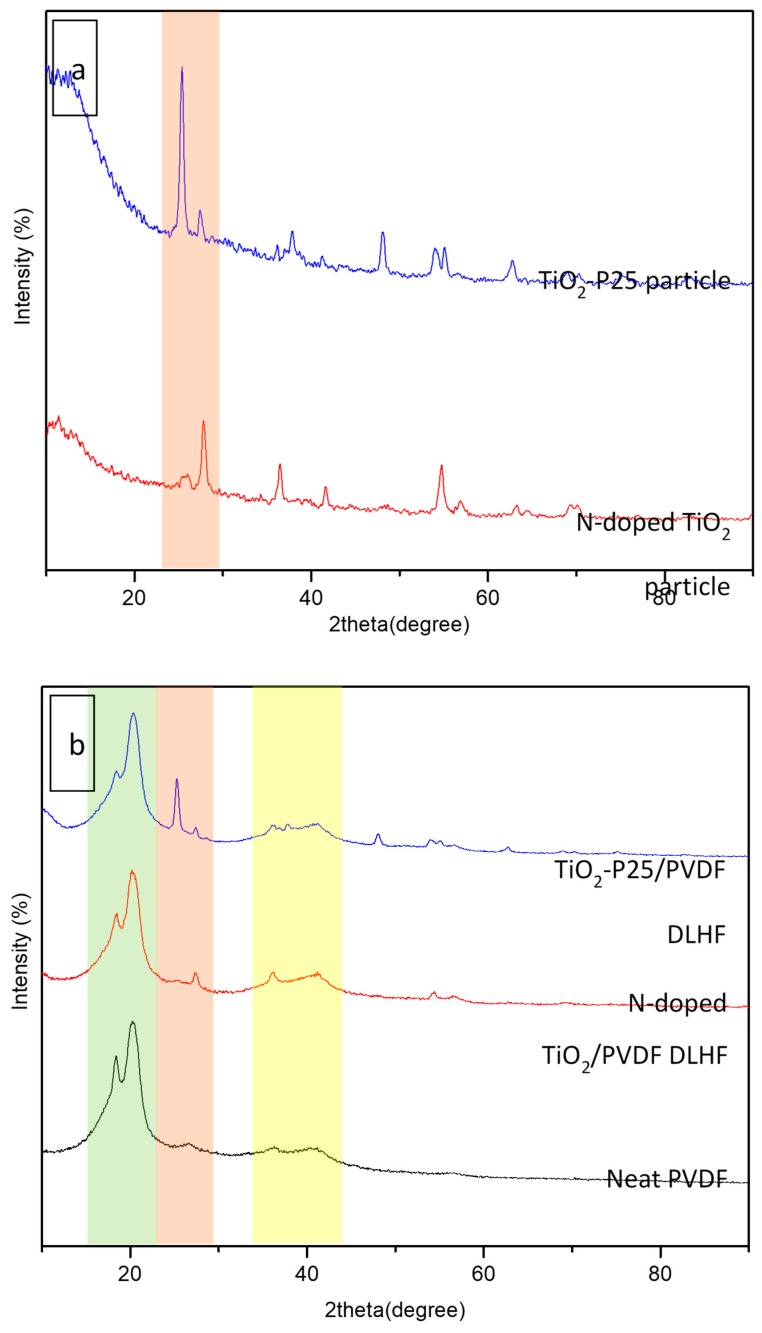
XRD pattern of (**a**) N-doped TiO_2_ particle and TiO_2_-P25 particle (**b**) N-doped TiO_2_/PVDF, TiO_2_-P25/PVDF DLHF and neat polyvinylidene fluoride (PVDF) membranes carried out at 40 kV and 40 mA, which employed a CuKα radiation at a wavelength of 0.15418 nm at an angular incidence of 2θ = 20–80°.

**Figure 4 membranes-10-00032-f004:**
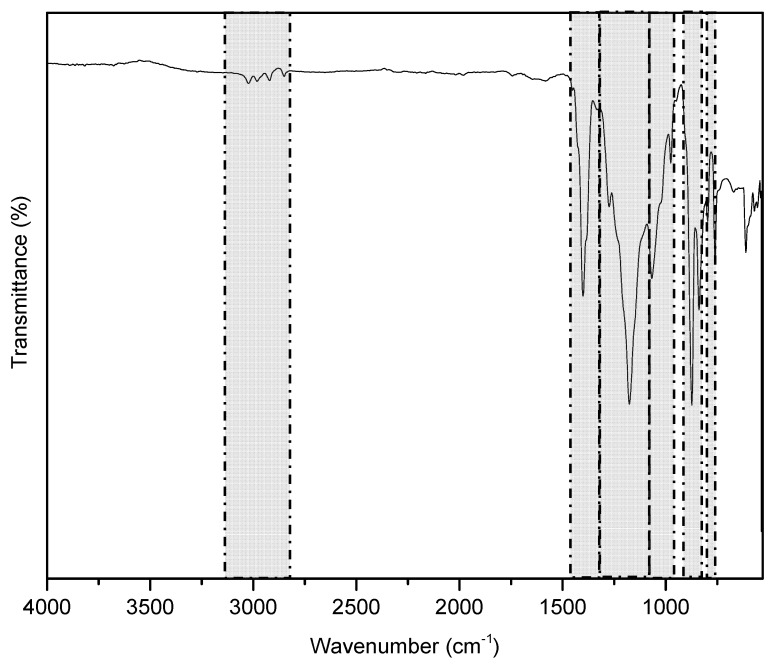
FTIR spectra of N-doped TiO_2_/PVDF DLHF membranes scanned with the wave number ranging from 650 to 4000 cm^−1^.

**Figure 5 membranes-10-00032-f005:**
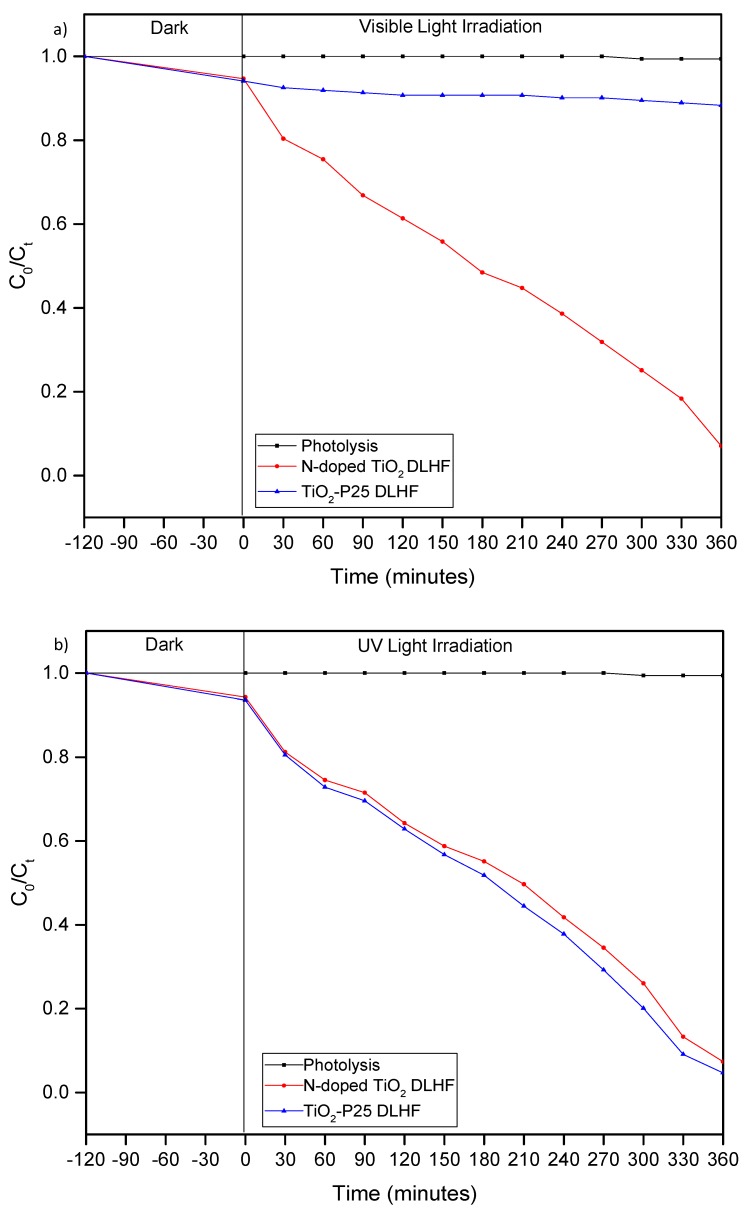
Photocatalytic degradation of BPA by N-doped TiO_2_ and TiO_2_-P25 DLHF membrane under (**a**) visible light and (**b**) UV light irradiation for 360 min of activity.

**Figure 6 membranes-10-00032-f006:**
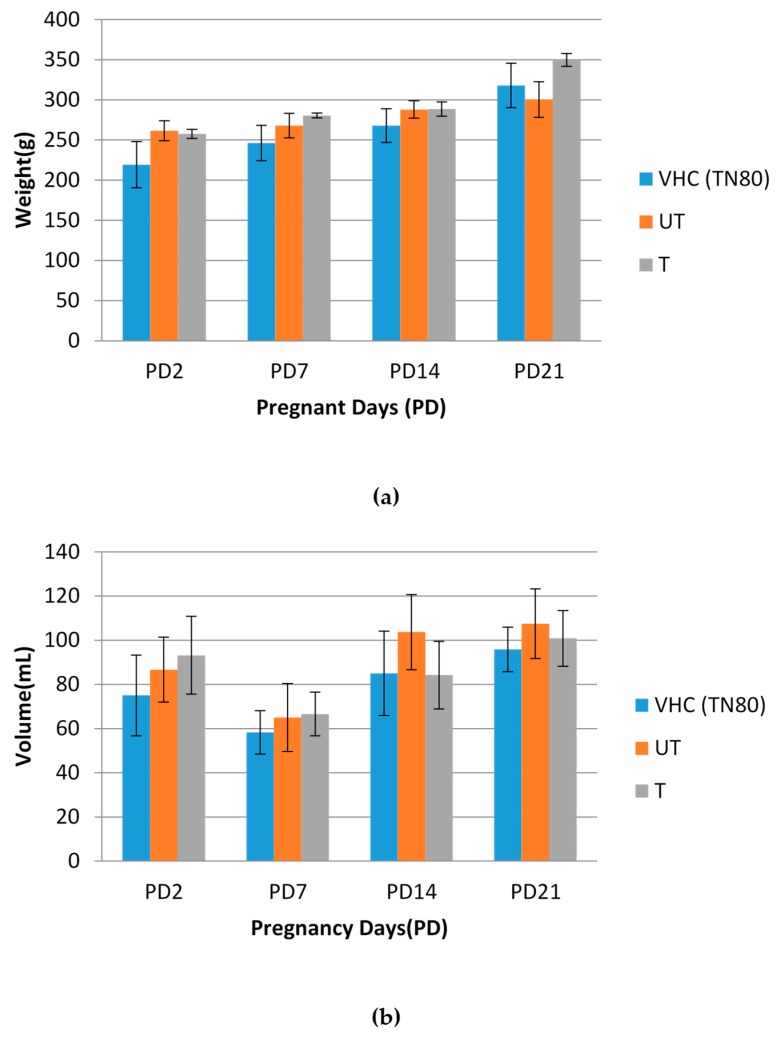
Effects of BPA on general health status of rats: (**a**) body weight of rats (wt); (**b**) drinking pattern of rats. Data are expressed as means. (*n* = 4–6 rats per group). VHC (TN80): Vehicle Control Group Tween 80; UT: Untreated BPA water; T: Treated BPA water.

**Figure 7 membranes-10-00032-f007:**
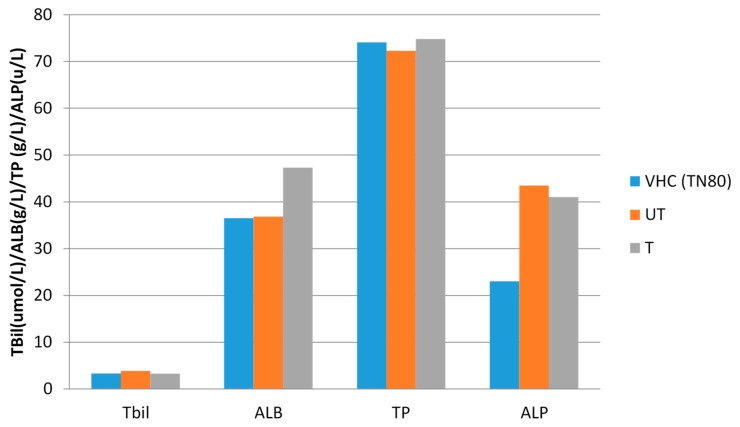
The liver function test results of treated rats. TBil: Total Bilirubin; ALB: Albumin; TP: Total Protein; ALP: Alkaline Phosphatase. VHC (TN80): Vehicle Control Group Tween 80; UT: Untreated BPA water; T: Treated BPA water.

**Figure 8 membranes-10-00032-f008:**
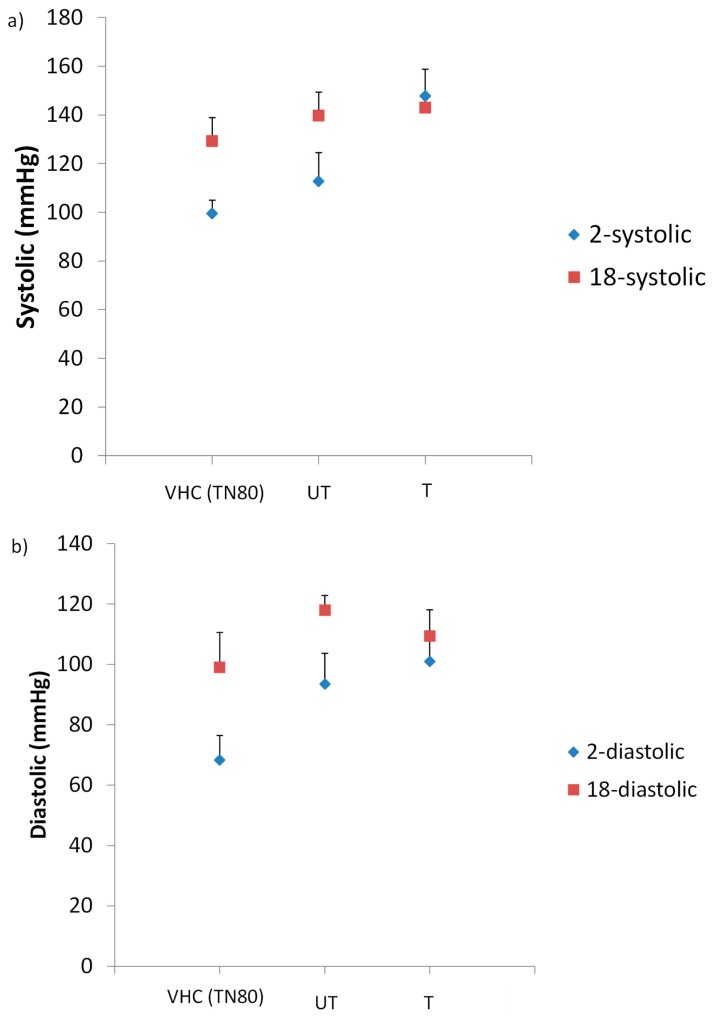
(**a**) Systolic and (**b**) diastolic blood pressure readings among treated rats. Data are expressed as means. (n ≥ 3 rats per group). 2-systolic/diastolic: diastolic BP reading on first week of treatment; 18-systolic/diastolic: diastolic BP reading on third week of treatment.

**Figure 9 membranes-10-00032-f009:**
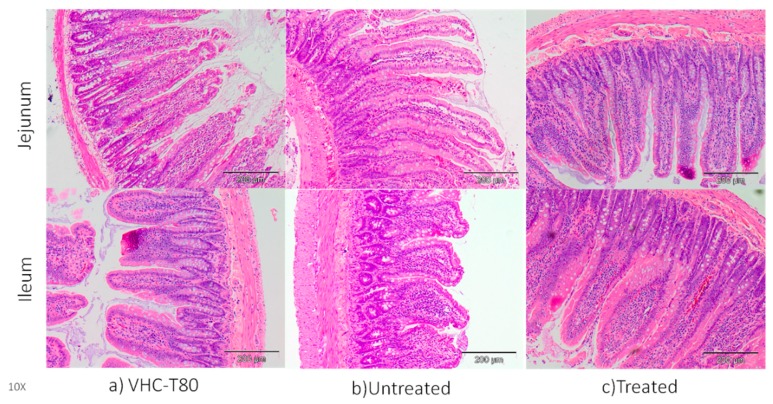
Representative diagram of H&E staining (10× magnification) of ileum and jejunum collected from rats (**a**) VHC (Tween 80) (**b**) Untreated (**c**) Treated BPA water and BPA water (*n* = 3–5).

**Table 1 membranes-10-00032-t001:** Dope compositions and spinning conditions of the dual-layer hollow fiber (DLHF) membranes [19,20,21].

**DLHF Configuration**	**N-Doped TiO_2_/TiO_2_ P25 DLHF**
Outer dope solution (wt%)	PVDF/N-doped TiO_2_ or TiO_2_ P25/DMAc (15/7.5/77.5)
Inner dope solution (wt%)	PVDF/PEG/DMAc (18/5/77)
	**Spinning Condition**
Outer dope flowrate (mL/min)	2
Inner dope flowrate (rpm)	26
Bore fluid	Distilled water
Bore fluid flow rate (mL/min)	8
Air gap (cm)	10
Take up speed (rpm)	5
Spinneret dimension (mm)	0.8/1.2/2.6
Outer dope flowrate (mL/min)	2

**Table 2 membranes-10-00032-t002:** Performance of N-doped TiO_2_/PVDF DLHF in comparison with TiO_2_-P25/PVDF DLHF membrane.

Configurations	N-Doped TiO_2_/PVDF DLHF	TiO_2_-P25/PVDF DLHF
Tensile Strength (MPa)	13.3 ± 0.24	14.5 ± 1.54
Elongation at Break (%)	189.2 ± 4.43	223.02 ± 9.68
Contact Angle (°)	70.4	72.6
Porosity	35.1	37.9
Water Flux (L/m^2^·h)	59.90	67.19
Band Gap (eV)	2.64	2.9

## Data Availability

The raw/processed data required to reproduce these findings cannot be shared at this time as the data also forms part of an ongoing study.

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
