# Peer review of "Visible-Light Active Photocatalytic Dual Layer Hollow Fiber (DLHF) Membrane and Its Potential in Mitigating the Detrimental Effects of Bisphenol A in Water"

_membranes, 2020, doi:10.3390/membranes10020032_

Round 1
Reviewer 1 Report
Manuscript entitled “Visible-light active photocatalytic dual layer hollow fiber (dlhf) membrane and its potential in mitigating the detrimental effects of bisphenol a in water” submitted by Roziana Kamaludin, Zatilfarihiah Rasdi, Mohd Hafiz Dzarfan Othman, Siti Hamimah Sheikh Abdul Kadir, Noor Shafina Mohd Nor, Jesmine Khan, Wan Nor I’zzah Wan Mohamad Zain, Ahmad Fauzi Ismail, Mukhlis A Rahman, Juhana Jaafar, can be accepted for publishing in the Membranes Journal in this form.
In this study is examined the efficiency of visible light driven photocatalytic dual-layer hollow fibre (DLHF) membrane for the removal of BPA from water and further investigated its detrimental effects by using an in-vivo model. The objectives of this study are presented since the beginning and are followed consistently throughout the manuscript. Each of experimental parameter is clearly presented, according with the experimental procedure, and detailed discussed in the manuscript. The manuscript presents original results obtained in a very well organized and systematic way, and the obtained results are consistent. In my opinion, this manuscript should be published in your Journal in this form.
Reviewer 2 Report
In this manuscript, the authors report on the photocatalytic degradation of BPA in water using membranes functionalized with doped titania nanoparticles. Typical TiO2 materials require UV light to drive their catalytic function, so many research groups have explored ways to sensitize TiO2 to lower energy photons. Nitrogen doping is one such strategy. Doped TiO2 membranes have been explored for various photocatalytic studies, but the specific degradation of BPA is a novel contribution. I am not qualified to referee the in vivo aspects of this work, so herein I focus on the materials/catalysis aspects only. In this context, the work warrants publication, pending revision to address the following minor issues:
The idea of integration catalysts into membranes to simultaneously perform a separation and a degradation is often discussed in the literature. However, an aspect that is frequently overlooked is the fact that the timescales for filtration and catalysis are generally quite different. This is indeed the case here. While a filtration occurs nearly instantaneously, catalysis often requires minutes or hours to degrade a significant amount of a target. Therefore, it is not actually true that one can perform both functions simultaneously. This nuance should be mentioned and discussed in the manuscript. Regarding modification of membranes with catalysts, there is a large literature that is overlooked here. Citing a review or two would help capture some of these (e.g. J. Appl. Phys. 124 (2018) 030901). Specifically on the topic of N-doped TiO2 for photocatalytic membranes, a missing reference is: Adv. Sust. Sys. 1 (2017) 1600041. An important topic that is overlooked in this work is the potential for unintentional degradation of the membrane's polymer support structure by catalytic action. Reactive oxygen species created by the N-TiO2 will surely lead to chemical reactions within the PVDF in addition to the intended chemistry of BPA. This will eventually damage the membrane and will limit its operational lifetime. Ideally, a long-time exposure experiment of pure water flux under illumination would characterize this phenomenon (by observing increasing flux over time). At a minimum, the issue should be discussed in the text.
Reviewer 3 Report
The manuscript by Kamaludin and co-workers describes the fabrication and performance tests of BPA removal membranes, complemented with in-vivo studies. The results are of interest to a broad audience and the topic fits well within the scope of the journal. However, there are several minor and major points to be addressed before a final decision can be reached.
1, The long paragraph covering references 1-18 should be significantly shortened as the detailed discussion on the biological effects of BPA is unnecessary for understanding the article. Simple mention a few reviews or book chapters on the topic.
2, 5 ppm BPA concentration was selected for the study. What is the basis for this value? A reference should be provided to support that this is a practically relevant concentration for potential application of the membranes for BPA removal from environmental samples.
3, The authors report some errors, for instance in Table 2, but their derivation is not clear. Are those standard deviations? What was the sample size? Were independently prepared membranes used for the measurements, or the same batch but different pieces were used? Clarify these points in the manuscript.
4, Some chemical characterization for the membranes should be provided, at least FTIR.
5, The grade and/or purity of all chemicals should be listed under section 2.1.1 within the description of the materials.
6, The authors mention the immobilization of catalysts into membrane matrix. Some recent diverse examples of this expanding field should be briefly mentioned (DOIs 10.1021/acscatal.8b01706; 10.1039/C8PY01789A; 10.1016/j.memsci.2019.117485); introduction line 61.
7, The figure and table captions are relatively short. Elaborate on the description so that the figures and tables stand on their own, which will help the readers to quickly navigate and understand the manuscript.
8, What is the expected lifetime and catalytic activity of the membranes? Reusability and recovery should be discussed.
9, Evidence should be provided for the strong interfacial interaction between layers.
10, The actual values of tensile strength and elongation at break should be briefly compared with other types of membrane materials to place the new membranes in a context and see how they differ (DOIs 10.1016/j.memsci.2015.12.057; 10.1021/acssuschemeng.9b02516; 10.1016/j.memsci.2019.117512).
11, The volumes for the soaking and washing of the membranes should be reported as L/m2 under the experimental section of the manuscript.
12, The manuscript needs to go through a thorough proofreading and editing as there are multiple typos and grammar mistakes throughout the text. Either a native speaker or an editing service should be utilized to improve the manuscript.
13, Follow the journal’s guideline on editing the reference list, it has inconsistencies and typos, do not use the et al abbreviation, and add the volume and page numbers everywhere.
Round 2
Reviewer 3 Report
The authors have done a thorough revision and the scientific aspects have significantly improved. However, there are some minor issues to be addressed:
1, Some figures, e.g. Figure 6-8, need to be appropriately edited. The resolution of text is poor, tick marks are often missing etc. To facilitate understanding the work, these issues need to be corrected.
2, The two panels of Figure 8 only has few datapoints and therefore the two panels can be merged.
3, The previous comment #6 was not address, the authors need to get back to this.
4, The reference list needs to be proofread. There are multiple typos, page numbers are missing, journal titles are abbreviated in some placed and not in others etc. Follow a consistent style throughout and complete the missing bibliographic information.
